# Enhanced Efficiency of MAPbI_3_ Perovskite Solar Cells with FAPbX_3_ Perovskite Quantum Dots

**DOI:** 10.3390/nano9010121

**Published:** 2019-01-19

**Authors:** Lung-Chien Chen, Ching-Ho Tien, Zong-Liang Tseng, Jun-Hao Ruan

**Affiliations:** 1Department of Electro-Optical Engineering, National Taipei University of Technology, Taipei 10608, Taiwan; chtien@mail.ntut.edu.tw (C.-H.T.); bj031001@gmail.com (J.-H.R.); 2Department of Electronic Engineering and Organic Electronics Research Center, Ming Chi University of Technology, New Taipei City 24301, Taiwan

**Keywords:** solar cells, perovskite, quantum dots, MAPbI_3_, FAPbX_3_

## Abstract

We describe a method to enhance power conversion efficiency (PCE) of MAPbI_3_ perovskite solar cell by inserting a FAPbX_3_ perovskite quantum dots (QD-FAPbX_3_) layer. The MAPbI_3_ and QD-FAPbX_3_ layers were prepared using a simple, rapid spin-coating method in a nitrogen-filled glove box. The solar cell structure consists of ITO/PEDOT:PSS/MAPbI_3_/QD-FAPbX_3_/C_60_/Ag, where PEDOT:PSS, MAPbI_3_, QD-FAPbX_3_, and C_60_ were used as the hole transport layer, light-absorbing layer, absorption enhance layer, and electron transport layer, respectively. The MAPbI_3_/QD-FAPbX_3_ solar cells exhibit a PCE of 7.59%, an open circuit voltage (Voc) of 0.9 V, a short-circuit current density (Jsc) of 17.4 mA/cm^2^, and a fill factor (FF) of 48.6%, respectively.

## 1. Introduction

Hybrid organic-inorganic perovskite materials have been widely accepted and applied in solar cells. The power conversion efficiency has evolved from 3.8% in 2009 to a certified value of 22.7% in 2018 [1,2,3,4,5]. Perovskite materials are a versatile material with unique optoelectronic properties, exhibiting strong light absorption, long diffusion length, and high mobility [6]. Developing high quality perovskite films is critical to improving perovskite solar cell efficiency. It is well known that high-performance solar cells originate from a low absorption bandgap (the optimal absorption band gap around a value of 1.1–1.5 eV) upon solar spectrum [7,8,9], low exciton binding energy [10,11], and long carrier diffusion length [12]. In addition, perovskite solar cells offer additional characteristics, like thin-film, flexibility, semitransparency, lightweight, and low-costs processing. A significant number of investigations have been focused on perovskite quantum dots (QDs) due to relatively abundant perovskite component sources [13,14,15]. The perovskite QD material present several advantageous properties, including band engineering through size and composition control, high absorption coefficient, light-response ranges (light absorption over a wide range of wavelengths from UV-visible to near IR), multiple exciton generation, cost-effectiveness, and solution process ability. Consequently, they are regarded as good light harvesters in perovskite solar cells [16,17,18,19,20]. While perovskite solar cells have experienced a steep increase in performance efficiency, they also show great potential to become a low-cost alternative to conventional solar cells. Due to the limited bandgap tunability of the perovskite material, near-infrared photons cannot be effectively captured in standard perovskite absorber layers by perovskite solar cells. We report on a two-step spin-coating perovskite solar cell manufacturing process, in which the MAPbI_3_/QD-FAPbX_3_ perovskite architecture is formed while using solvent-engineering techniques. Besides, the MAPbI_3_/QD-FAPbX_3_ layer exhibits a marginally broad light absorption region; the current density is also prominently enhanced, which is beneficial for improving the power conversion efficiency. These results explain why perovskite solar cells containing FAPbX_3_ QDs exhibit a much higher PCE than cells without.

## 2. Materials and Methods 

The 0.2 m mol of FAPbX_3_ (X = I, Br) (FAPbBr_1.5_I_1.5_ and FAPbBrI_2_) perovskite QD solution were prepared in this study while using a simple and rapid method. Table 1 shows the FAI (Lumtec Corp., Taipei, Taiwan), FAB (Lumtec Corp., Taipei, Taiwan), PbI_2_ (Alfa Aesar, Lancashire, UK), and PbBr_2_ (Alfa Aesar, Lancashire, UK) perovskite FAPbX_3_ (X = I, Br) QDs solution parameters. The FAB powder, FAI powder, PBI_2_ powder, and PbBr_2_ powder were decanted into a solution of 2 mL of dimethylformamide (DMF) (Echo Chemical Co., Ltd., Miaoli, Taiwan); and, 300 µL of oleic acid (Echo Chemical Co., Ltd., Miaoli, Taiwan) and 5 µL of n-octylamine (Echo Chemical Co., Ltd., Miaoli, Taiwan) were decanted into this mixture to form the FAPbX_3_ (X = I, Br) precursor solution. Subsequently, A 20 µL of FAPbX_3_ (X = I, Br) precursor solution was decanted into 2 mL of chlorobenzene (Echo Chemical Co., Ltd., Miaoli, Taiwan) and 3 mL of ethyl acetate (Echo Chemical Co., Ltd., Miaoli, Taiwan), followed by the centrifugal process to separate the red precipitate from the FAPbX_3_ (X = I, Br) precursor solution. The said red precipitate was dried under vacuum for 12 h to entirely eliminate the solvent. The FAPbX_3_ (X = I, Br) powder was then dissolved in 60 µL of chlorobenzene to prepare the FAPbX_3_ (X = I, Br) perovskite QD solution.

The patterned ITO glass (Ruilong Corp., Miaoli, Taiwan) was cleaned using ultrasonic treatment in deionized (DI) water, acetone, and isopropanol. The ITO glass was then treated in a UV cleaner for 10 min. The cleaned ITO glass was coated with a PEDOT:PSS solution (UMAT Corp., Hsinchu, Taiwan) using a spin-coating method at 4000 rpm for 30 s, followed by heating at 120 °C for 10 min. The solution of the perovskite precursor was prepared using the 289 mg Pbl_2_ and 98 mg methylammonium iodide (MAI) (Lumtec Corp., Taipei, Taiwan) solvent in 500 µL of a dimethyl sulfoxide (DMSO) (Echo Chemical Co., Ltd., Miaoli, Taiwan) and DMF cosolvent with a volume ratio of 1:9 in a glove box that was filled with forming gas of nitrogen. Next, the solution of the perovskite precursor were spin-coated onto the PEDOT:PSS/ITO glass using two-step spin-coating processes with 1000 rpm and 5000 rpm for 10 s and 20 s, respectively. At the second coating step, the said spin-coated layer was suppressed by the dropping 100 µL of anhydrous toluene (Echo Chemical Co., Ltd., Miaoli, Taiwan). Subsequently, the spin-coating process was undertaken and the samples were annealed at temperature 100 °C for 5 min. Afterward, the FAPbX_3_ (X = I, Br) (FAPbBr_1.5_I_1.5_ and FAPbBrI_2_) QDs dispersed in chlorobenzene (60 µL) were spin coated onto the above MAPbI_3_ perovskite layer at 1000 rpm for 30 s, followed by standing for 2 min, respectively. Finally, a C_60_ layer (Uni-Onward Corp., New Taipei, Taiwan) and Ag electrode (UMAT Corp., Hsinchu, Taiwan) were thermally formed sequentially by a thermal evaporator method with a high-vacuum ambient of 1.5 × 10^−6^ torr to finish the device structure. The samples were shielded with a shadow mask during the C_60_/Ag forming, which define an active area of 0.5 × 0.2 cm^2^. Figure 1 schematically describes the complete structure. The PEDOT:PSS, MAPbI_3_, QD-FAPbX_3_, and C_60_ roles in the device structure are the hole transport layer, light-absorbing layer, absorption enhancement layer, and electron transport layer, respectively

## 3. Results and Discussion

Figure 2 shows the TEM (transmission electron microscopy) (Tecnai F30, Philips, Netherlands) images of the QD-FAPbBr_1.5_I_1.5_ and QD-FAPbBrI_2_ thin films. It is clear that the QD-FAPbBr_1.5_I_1.5_ and QD-FAPbBrI_2_ thin films are composed of many quantum dots, in which the quantum dot sizes ranged between 7 nm and 10 nm.

Figure 3 presents the photoluminescence (PL) spectra of the QD-FAPbBr_1.5_I_1.5_ and QD-FAPbBrI_2_ solution were obtained using a photoluminescence system (UniRAM, Protrustech). The QD-FAPbBr_1.5_I_1.5_ and QD-FAPbBrI_2_ peak locations are observed at 680 nm and 730 nm, respectively. This indicated that the band gap of QD-FAPbX_3_ can be expected to decrease when the proportion of PbI_2_ increases, such that it results in a red shift in the emission peak of the PL spectrum. That is, the observed red shift of the PL emission can be interpreted as a lowering of the band gap of the QD-FAPbX with a PbI_2_ proportion increase. Insets are pictures of the QD-FAPbBr_1.5_I_1.5_ and QD-FAPbBrI_2_ specimens that emitted near infrared light while being excited by a 405-nm laser.

Figure 4 plots the transmittance and absorbance spectra of MAPbI_3_, MAPbI_3_/QD-FAPbBr_1.5_I_1.5_, and MAPbI_3_/QD-FAPbBrI_2_ films measured using an UV/VIS/NIR spectrophotometer (U-4100, Hitachi). MAPbI_3_, MAPbI_3_/QD-FAPbBr_1.5_I_1.5_, and MAPbI_3_/QD-FAPbBrI_2_ films transmittances (see Figure 4a) were below 10% between 300 nm and 700 nm, but above 30% between 750 nm and 850 nm. In particular, the highest value of the transmittance spectrum was above 50% between 780 nm and 850 nm in the MAPbI_3_ film spectrum on ITO glass. The absorption edge location was at 750 nm, which corresponded to the MAPbI_3_ film absorption. Figure 4b shows the absorption spectra of the MAPbI_3_, MAPbI_3_/QD-FAPbBr_1.5_I_1.5_, and MAPbI_3_/QD-FAPbBrI_2_ films. The absorption spectra between 300 nm and 850 nm show that the MAPbI_3_/QD-FAPbBrI_2_ film was higher than that of MAPbI_3_ and MAPbI_3_/QD-FAPbBr_1.5_I_1.5_ films. One may observe that the MAPbI_3_/QD-FAPbBr_1.5_I_1.5_ and MAPbI_3_/QD-FAPbBrI_2_ films were absorbed with noteworthy between 780 nm and 850 nm, as compared to the MAPbI_3_ films. This can increase the spectral absorption of the solar cell, thereby generating more short-circuit current density and enhancing the power conversion efficiency.

The electrical J-V (current density-voltage) properties were obtained using a Keithley 2420 sourcemeter under irradiation by a 100 W xenon lamp. The irradiation condition on the sample surface was adjusted at a power density of 100 W/m^2^ (AM1.5). Figure 5 presents the relationship between the current density and the voltage (J-V) for MAPbI_3_, MAPbI_3_/QD-FAPbBr_1.5_I_1.5_, and MAPbI_3_/QD-FAPbBrI_2_ solar cells. Table 2 summaries the performance of the three different solar cells. The MAPbI_3_/QD-FAPbBrI_2_ optimal device demonstrated outstanding performance: power conversion efficiency (Eff) = 7.59%, short-circuit current density (Jsc) = 17.4 mA/cm^2^, open-circuit voltage (Voc) = 0.9 V, and fill factor (FF) = 48.6%. Clearly, the solar cell with the MAPbI_3_/QD-FAPbBrI_2_ perovskite absorber exhibited the greatest increase in absorption, ranging from 300 nm to 850 nm. The cells with the MAPbI_3_/QD-FAPbBrI_2_ and MAPbI_3_/QD-FAPbBr_1.5_I_1.5_ perovskite absorbers exhibited the improved external quantum efficiency (EQE), as shown in Figure 5b. The EQE curves were used to calculate the integration current of cells based on MAPbI_3_/QD-FAPbBrI_2_, MAPbI_3_/QD-FAPbBr_1.5_I_1.5_, and MAPbI_3_ were 16.2, 15.8, and 7.9 mA/cm^2^. These values are different from the Jsc values that were obtained from J-V curves, which may be due to the samples, has been stored in a N_2_--filled glove box over half a year. This means that the FAPbX_3_ perovskite quantum dots coated on MAPbI_3_ can increase the short-circuit current. This means that the MAPbI_3_/QD-FAPbBrI_2_ perovskite absorber can produce a high-efficiency thin film solar cell.

## 5. Conclusions

In conclusion, we demonstrated PCE enhancement of the MAPbI_3_ perovskite solar cells with FAPbX_3_ perovskite quantum dot via increasing the absorption range of the spectral. The PL spectra of the QD-FAPbX_3_ indicated a red shift when the proportion of PBI_2_ increases due to the band gap decrease. The absorption spectra showed that the MAPbI_3_/QD-FAPbX_3_ films exhibited broad spectral absorption. The device performance was optimized while using a MAPbI_3_/QD-FAPbBrI_2_ layer, with a short-circuit current density (J_SC_) = 17.4 mA/cm^2^, open-circuit voltage (V_OC_) = 0.9 V, fill factor (FF) = 48.6%, and power conversion efficiency (Eff) = 7.59%. The power conversion efficiency of the cell with the MAPbI_3_/QD-FAPbX_3_ film improved remarkably around 43.7% in the best case.

## Figures and Tables

**Figure 1 nanomaterials-09-00121-f001:**
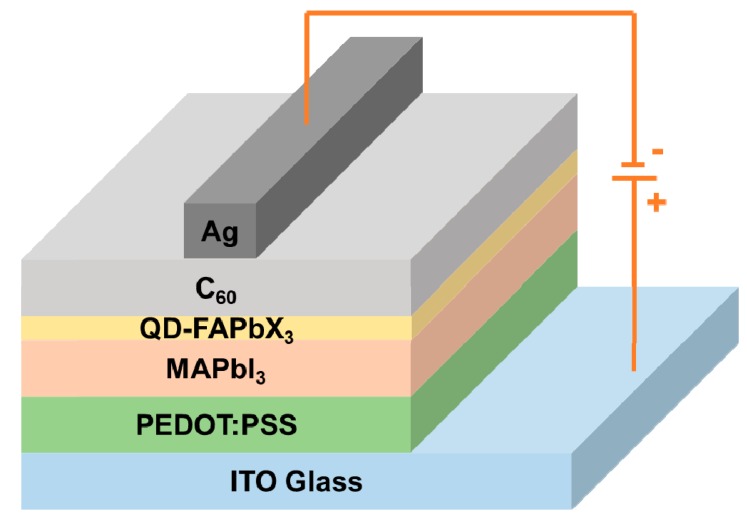
Schematic of the perovskite solar cell device configuration of a structure of ITO/PEDOT:PSS/MAPbI_3_/QD-FAPbX_3_/C_60_/Ag.

**Figure 2 nanomaterials-09-00121-f002:**
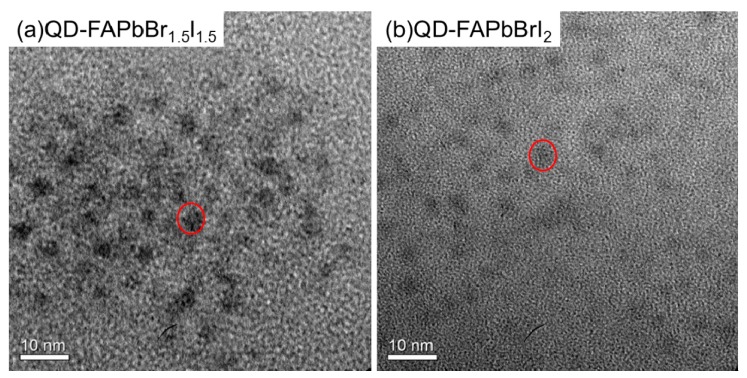
TEM morphological image of perovskite film. (**a**) QD-FAPbBr_1.5_I_1.5_ film and (**b**) QD-FAPbBrI_2_ film.

**Figure 3 nanomaterials-09-00121-f003:**
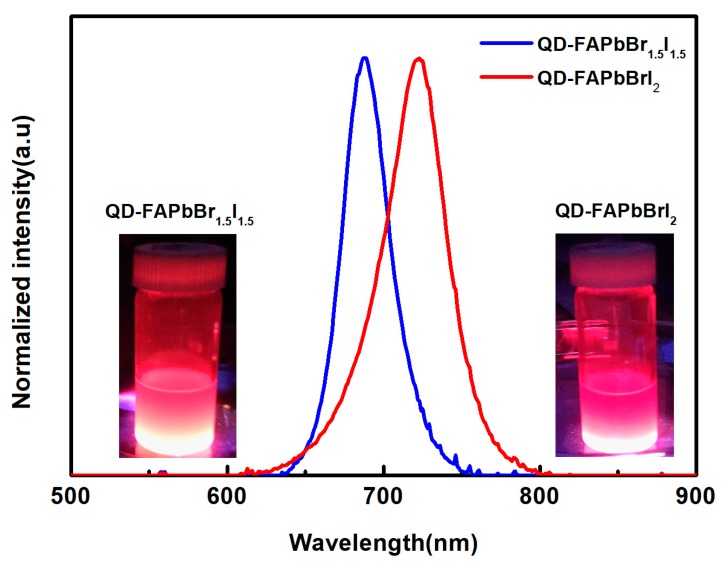
Room-temperature PL spectra of the QD-FAPbBr_1.5_I_1.5_ and QD-FAPbBrI_2_ solution. The inset shows the photograph of QD-FAPbBr_1.5_I_1.5_ and QD-FAPbBrI_2_ excited by a 405 nm laser.

**Figure 4 nanomaterials-09-00121-f004:**
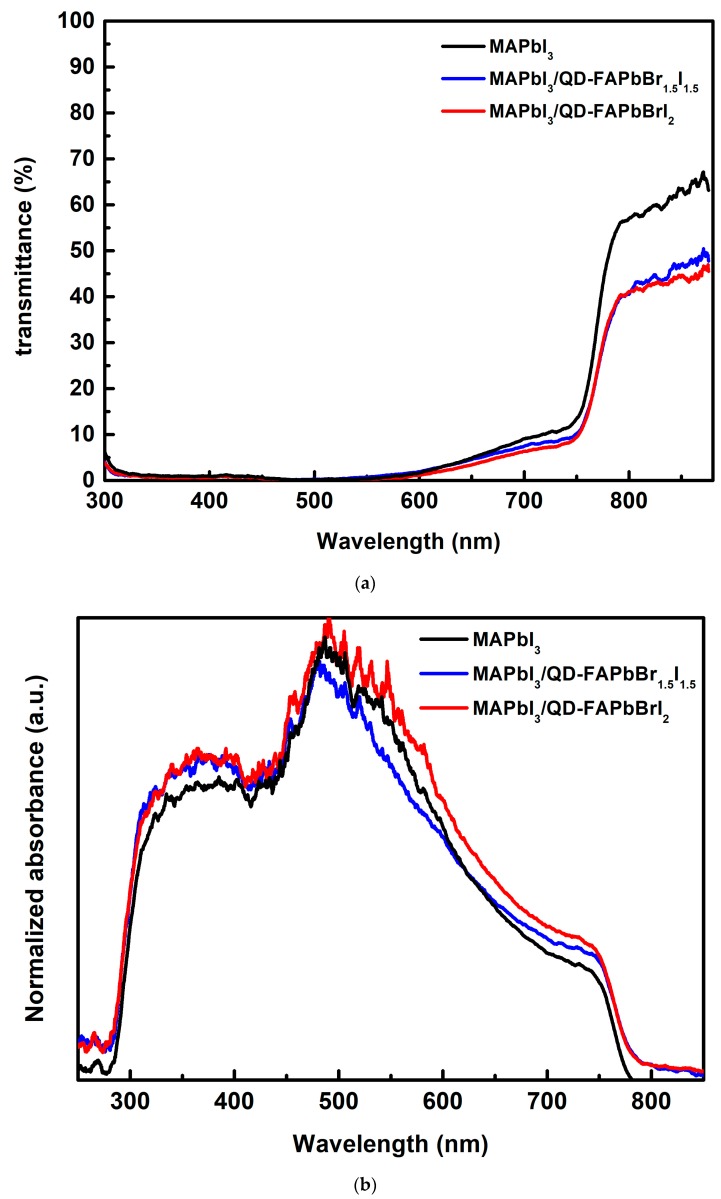
(**a**) Transmittance and (**b**) absorbance spectra of the MAPbI_3_, MAPbI_3_/QD-FAPbBr_1.5_I_1.5_, and MAPbI_3_/QD-FAPbBrI_2_ films.

**Figure 5 nanomaterials-09-00121-f005:**
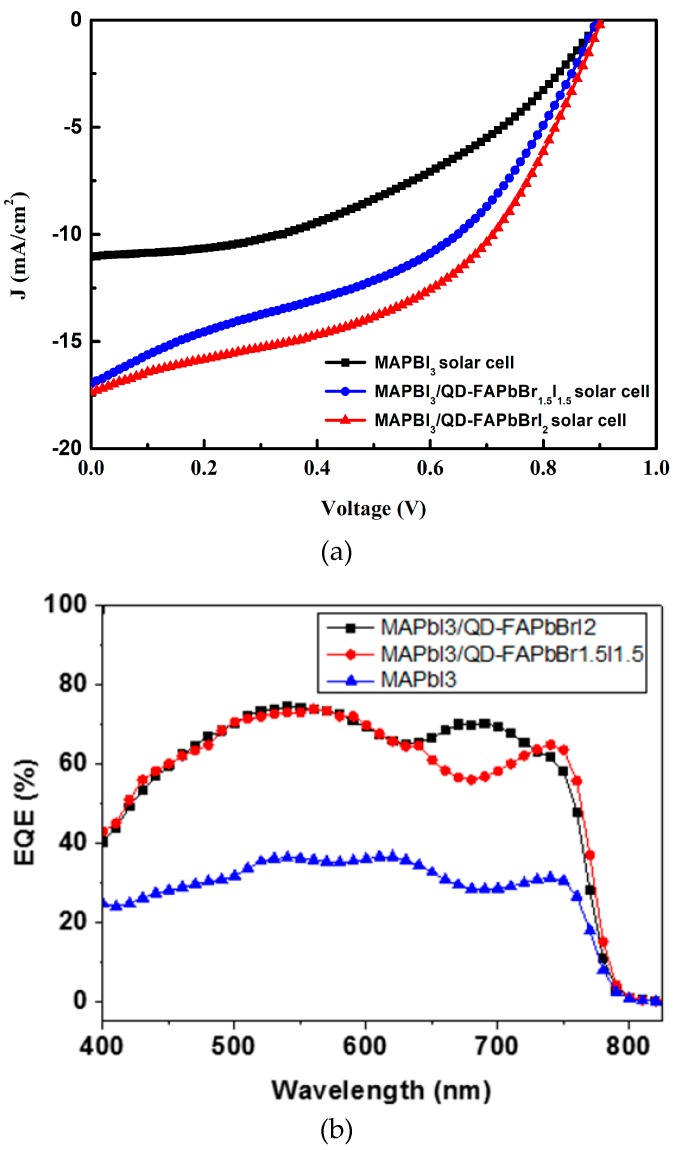
(**a**) Current density–voltage properties and (**b**) external quantum efficiency (EQE) spectra of perovskite solar cells.

**Table 1 nanomaterials-09-00121-t001:** Parameters of FAPbX_3_ (X = I, Br) perovskite quantum dots (QDs) solution.

Material	FAPbBr_1.5_I_1.5_	FAPbBrI_2_
FAI	17.1 mg	--
FAB	12.5 mg	25 mg
PbI_2_	46.1 mg	92.2 mg
PbBr_2_	36.7 mg	--

**Table 2 nanomaterials-09-00121-t002:** Solar cell photovoltaic parameters. The measurement was carried out under simulated one sun (100 mW/cm^2^).

Sample	Voc (V)	Jsc (mA/cm^2^)	FF (%)	Eff (%)
MAPbI_3_	0.9	11	53	4.27
MAPbI_3_/QD-FAPbBr_1.5_I_1.5_	0.89	17	43.3	6.54
MAPbI_3_/QD-FAPbBrI_2_	0.9	17.4	48.6	7.59

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
