# Peer review of "Enhanced Efficiency of MAPbI3 Perovskite Solar Cells with FAPbX3 Perovskite Quantum Dots"

_nanomaterials, 2019, doi:10.3390/nano9010121_

Round 1

Reviewer 1 Report

The english level is very low  (cf my corrected version).
The quality of some figures is far too low for any paper publication.
The overall quality of presentation of the paper is too low for publication in a scientific journal   but the scientifc content worths to be published.

I strongly advise the authors to submit their revised paper  after  sighnificant improvement  and carefull reading of the recent literature.  It seems to me that important paper  or communication are not reported here.
They should also find  interpretation about SWIR absorption (2 PA process seems to be enhenced in QDs). This would help to also to show  the novelty of the paper. I'm not a specialist of PV but it appears to me that several papers present QD use for emission  but fewer for  absorption an PV application.

The conclusion should present some perspective of improvement.
Sincerely.   

Author Response

The auther's response PDF file is attached.

Reviewer 2 Report

The present manuscript entitled "Enhance efficiency of MAPbI3 perovskite solar cells with FAPbX3 perovskite quantum dots'' describes the fabrication of a bilayer perovskite solar cell upon the incorporation of a FAPbX3 perovskite quantum dots layer onto the main MAPbI3 layer and the efficiency enhancement achieved.

This demonstrated concept is interesting. However, I do not recommend the present manuscript for publication to Nanomaterials in its current form, since it contains serious flaws as described below. The authors’ assertions are not well confirmed, since important characterization tools are missing, as well as the novelty of the manuscript is not pointed out. Therefore, I recommend its resubmission after major revision.

My comments are below:

1) All PV parameters with the exception of the Voc are much lower than the SOA. Why?

2) XRD patterns of the perovskites are not provided.

4) What is the average grain size of the perovskites? A grain size distribution for the obtained grains in the presence and without the FAPbX3 dot layer should be provided.

5) Standard deviations should be inserted in Table 2. In addition, which is the active area of the device?

6) PV parameters for both forward and reverse scans should be provided.

7) EQE is not provided.

8) Any comment about the devices’ stability?

Author Response

Response to Editor and Reviewer Comments

(Manuscript ID: nanomaterials-397712)

We appreciate the comments and suggestions from the editor and reviewers, and the manuscript has been revised. In the revised manuscript, the added and modified texts are shown in red words. The detailed changes can be seen in the attachment.

Reviewer 3 Report

Within the manuscript the authors reported on a two-step spin-coating perovskite solar cell manufacturing process, in which the MAPbI3/QD-FAPbX3 perovskite architecture is formed using solvent-engineering techniques.

The manuscript is clear, well arranged, the experimental and theoretical results are well presented. The research is scientifically sound, and the motivation is clear.

I have only some comments to do:

In my opinion the idea is quite interesting but the results (PCE values) are not impressive as much higher efficiencies of around 20% are usually reported [see 729. Perovskites-Based Solar Cells: A Review of Recent Progress, Materials and Processing Methods Zhengqi Shi and Ahalapitiya H. Jayatissa Materials 2018 May; 11(5): and references therein]

Therefore it needs much more work in order to be “competitive”.

Besides this:

- It would be very useful to show a diffraction pattern of the QD-FAPbX3 perovskite quantum dots and of the QD-FAPbX3 layer to show that the QD structure has preserved

 - It would be very useful to show the PL spectra on MAPbI3, MAPbI3/QD-FAPbBr1.5I1.5, and 1 MAPbI3/QD-FAPbBrI2 films. 2 by similarity with the Figure 4

Author Response

Response to Editor and Reviewer Comments

(Manuscript ID: nanomaterials-397712)

We appreciate the comments and suggestions from the editor and reviewers, and the manuscript has been revised. In the revised manuscript, the added and modified texts are shown in red words. The detailed changes are described as follows.

Reviewer:
Comments to the Author:
1.
In my opinion the idea is quite interesting but the results (PCE values) are not impressive as much higher efficiencies of around 20% are usually reported [see 729. Perovskites-Based Solar Cells: A Review of Recent Progress, Materials and Processing Methods Zhengqi Shi and Ahalapitiya H. Jayatissa Materials 2018 May; 11(5): and references therein]

Therefore it needs much more work in order to be “competitive”.

Response:

In our study, after the solar cell is added to the QD-FAPbBrI2 film, the power conversion efficiency (PCE) is improved by absorbing a longer wavelength range. Part of the QD-FAPbX3 agglomerates to form an island shape due to the excessive concentration of chlorobenzene as a dispersing solvent. On the other hand, QD-FAPbX3 film has a low FF value due to poor flatness, which affects PCE efficiency.

2. It would be very useful to show a diffraction pattern of the QD-FAPbX3 perovskite quantum dots and of the QD-FAPbX3 layer to show that the QD structure has preserved.

Response:

No peaks can be detected in all QD layers because of thin thickness. So QD layers/MAPbI3 show only MAPbI3 peaks. The XRD patterns of the MAPbI3 films are similar with our previous reports. 

3. It would be very useful to show the PL spectra on MAPbI3, MAPbI3/QD-FAPbBr1.5I1.5, and 1 MAPbI3/QD-FAPbBrI2 films. 2 by similarity with the Figure 4.

Response:

The suspend QD was used in Figure 4, so the PL spectra can be seen clearly. Once QD layer formed on the MAPbI3 films, only PL of MAPbI3 can be identified because of too thin thickness of QD layers.

Round 2

Reviewer 1 Report

Overall remarks.

This paper results worth publication. The quality of figures has been improved. The English level of the paper as increased also but  still needs efforts for publication  to my point of view.

It seems that authors have only partially read the reviewer’s reports as important grammatical mistakes have not been corrected.  

Author Response

Thank you for your comments. The manuscript has revised according to your suggestion.

Reviewer 2 Report

The manuscript is significantly improved upon the authors' revision. However, EQE is very important to be performed, since it confirms the extracted results from the J-V curves. I do not recommend this study for publication in its current form.

Author Response

Thank you for your comments. I am sorry that the equipment is hard to book right now. However, there are the absorbance spectra and J-V curves in the manuscript to show the effect of QD-FAPbX3.

Reviewer 3 Report

As I said, the results (PCE values) are not impressive as much higher efficiencies of around 20% are usually reported and I do not see a convincing comment to this remark. The authors has to present/discuss (briefly) some perspective for the PCE improvement in a convincing manner otherwise their study lacks impact within the community despite its novelty.

Author Response

Thank you for your comments. However, please mind the purpose of this paper is to discuss the impact of QD-FAPX on the characteristics of MAPbI3 solar cells.